# Assessing Health and Economic Benefits of Omega-3 Fatty Acid Supplementation on Cardiovascular Disease in the Republic of Korea

**DOI:** 10.3390/healthcare11162365

**Published:** 2023-08-21

**Authors:** Moon Seong Kim, Jin Man Kim, Sang Gyeong Lee, Eun Jin Jung, Sang Hoon Lee, Wen Yan Huang, Bok Kyung Han, Da Eun Jung, Sung Bum Yang, Inbae Ji, Young Jun Kim, Ji Youn Hong

**Affiliations:** 1Department of Food and Regulatory Science, Korea University, Sejong 30019, Republic of Korea; tjdans44@korea.ac.kr (M.S.K.); wlsaks85@korea.ac.kr (J.M.K.); lee30@korea.ac.kr (S.G.L.); ejjung1124@korea.ac.kr (E.J.J.); hanmoo@korea.ac.kr (B.K.H.); 2Department of Food and Biotechnology, Korea University, Sejong 30019, Republic of Korea; t9592359@korea.ac.kr (S.H.L.); flyhighwy@korea.ac.kr (W.Y.H.); 3BK21 FOUR Research Education Team for Omics-Based Bio-Health in Food Industry, Korea University, Sejong 30019, Republic of Korea; 4Department of Environmental and Resource Economics, Dankook University, Chungnam 16890, Republic of Korea; smiler@dankook.ac.kr (D.E.J.); passion@dankook.ac.kr (S.B.Y.); 5Department of Food Industrial Management, Dongguk University, Seoul 04620, Republic of Korea; jiinbae@dongguk.edu

**Keywords:** omega-3, cardiovascular disease, meta-analysis, healthcare cast, cost saving analysis

## Abstract

Background: Cardiovascular disease (CVD) is the primary cause of mortality worldwide and imposes a significant social burden on many countries. Methods: This study assessed the health and economic benefits of omega-3 associated with CVD. The meta-analysis estimated the risk ratio (RR) and absolute risk reduction (ARR), and the economic impact was calculated using direct and indirect costs related to CVD treatments in Korean adults. Results: A total of 33 studies were included in the meta-analysis on CVD outcomes, with 80,426 participants in the intervention group and 80,251 participants in the control group. The meta-analysis determined a significant reduction in omega-3 in CVD (RR = 0.92, 95% CI: 0.86~0.97) and ARR (1.48%). Additionally, the subgroup analysis indicated that higher doses and the long-term consumption of omega-3 could further enhance these effects. After applying ARR from meta-analysis to the target population of about 1,167,370 in 2021, the Republic of Korea, it was estimated that omega-3 consumption could result in an economic benefit of KRW 300 billion by subtracting the purchase expenses of omega-3 supplements from the total social cost savings. Conclusion: Omega-3 supplements can help to reduce the risk of CVD and subsequent economic benefits in the Republic of Korea.

## 1. Introduction

Cardiovascular disease (CVD) is among the leading causes of mortality worldwide, even with advances and efforts in disease prevention and treatment [1]. The World Health Organization (WHO) estimated that 17.9 million people died from CVD in 2019, representing 32% of all global deaths. Of these deaths, 85% were due to a heart attack and stroke [2]. In the Republic of Korea, among the major chronic diseases excluding cancer, the highest mortality rate was attributed to CVD, with 61.5 deaths per 100,000 in the population in 2021. The Korea Disease Control and Prevention Agency reported that the socio-economic burden of CVD has been rapidly increasing [3]. Since chronic degenerative diseases, including CVD, drive healthcare costs to an alarming annual rate, Korean health policies have tried to prioritize the prevention of CVD-based morbidity and mortality.

Eicosapentaenoic (EPA) and docosahexaenoic (DHA) acids, the two primary omega-3 polyunsaturated fatty acids of marine origin, have shown a probability for preventing CVD outcomes in many epidemiologic studies [4]. Recognizing the benefits of omega-3 fatty acids, many governments and organizations recommend consuming EPA and DHA to prevent CVD, including foods, healthy functional foods, or supplements. For example, the United Kingdom’s Scientific Advisory Committee established dietary recommendations for adults in the general population to intake at least 450 mg EPA+DHA per day [5] and 400–500 mg EPA+DHA per day with at least 100–120 mg DHA per day set by the French Agency [6]. At the same time, the Food and Agricultural Organization has recommended a minimum of 250 mg EPA+DHA per day as the appropriate daily intake [7].

Various studies on omega-3 supplementation have been conducted as part of efforts to prevent CVDs, and large-scale RCT studies on the effects of omega-3 intake, including REDUCE-IT, GISSI, STRENGTH, and JELIS, have been conducted [8,9,10,11]. However, the benefits of omega-3 supplementation in reducing the risk of cardiovascular disease are still controversial due to the diversity of CVD [12]. In addition, the results of meta-analyses for the effect of EPA and DHA on multiple CVD outcomes have shown equal inconsistencies; for example, Abdelhamid et al. reported a low certainty of a possible protective effect against coronary heart disease (CHD) mortality [13], while Maki et al. found a statistically significant 8.0% risk reduction [14]. The reasons for this variability among the results of RCTs may reflect differences in deriving a comprehensive meta-analysis using the different definitions of major adverse cardiovascular events (MACE) by each author to obtain integrated results for various CVD. A previous study by Galan et al. reported that MACE includes non-fatal myocardial infarction (MI), stroke, and CVD death [15], while Nicholls’ study added emergent coronary revascularization and hospitalization for unstable angina [10]. In addition, the results of the meta-analysis depend on various contents, including the range of publication years, the criteria for the inclusion of research contents, the criteria for subjects, and so on [16,17]. Since research results of omega-3’s effect on CVD are continuously being published, this study attempted to include studies up to the latest data as much as possible with a more extensive evaluation of the evidence and transparent criteria.

Omega-3 supplementation has been reported to improve health and reduce total costs under various scenarios, including the Markov simulation, incremental accounting life years, and clinical probabilities calculating [18,19,20,21], although each study has theoretical limitations. Research results measured using the Congressional Budget Office (CBO) accounting methods in the United States (US) reported that giving each member of the Medicare program approximately 1800 mg of omega-3 fatty acids per day could reduce hospital and physician costs by USD 3.2 billion over five years [18]. It concluded that omega-3 supplementation could be implicated in fewer fatal myocardial infarctions, with less cardiovascular mortality and cost savings compared to no supplementation in the US.

Alternatively, a series of studies by Frost and Sullivan utilized meta-analysis-estimated risk ratio (RR) reductions in disease risks to calculate healthcare cost savings and prevent cardiovascular disease through an omega-3 intervention in Australia, the European Union (EU), and the US [22,23,24]. In the 2014 report for Australia, the average annual net economic benefit during 2015 and 2020 was estimated to be USD 530 million, and the average annual benefit/cost ratio was USD 8.49 per USD 1 spent on omega-3 regimens on CVD [22]. For the EU, the impact of omega-3 use represented a relative risk for an individual in the target population experiencing a CVD-attributed adverse event to be reduced by 4.9% given the daily use of 1000 mg of omega-3 EPA and DHA food supplements [23]. This result corresponds to over 1.5 million avoided CVD-attributed hospital events throughout the EU over the next five years. If 100% of the EU target population used 1000 mg omega-3 EPA and DHA food supplements daily, the net avoidable CVD-attributed costs per year could be realized at EUR 188 per person, and the benefit/cost ratio (avoided CVD-attributed costs per EUR 1 spent on Omega-3) would be EUR 2.29. The report on the US Health Care Cost Savings from the Targeted Use of Dietary Supplements revealed that the net cost savings expected from reduced healthcare expenditures as a result of avoided CVD-related events via omega-3 consumption would reach USD 3.70 billion in 2022 or USD 4.47 billion per year in net savings from 2022 to 2030 [24].

In the Republic of Korea, oils containing EAP and DHA are registered as functional materials and are approved officially based on the Health Functional Foods Act for various improvements such as blood triglycerides, blood circulation, memory, and eye health [25], and consumption of omega-3 supplements has been steadily increasing. Nevertheless, few studies have been conducted on the healthcare cost associated with cardiovascular disease from omega-3 consumption in the Republic of Korea. A study by Hwang et al. estimated that the net medical cost savings caused by a CVD incidence reduction due to the omega-3 intake in the Korean elderly group was about KRW 210 billion over seven years from 2005 to 2011; however, they used an average annual cost for the daily intake of omega-3 from US data [26]. Therefore, further studies still need to be adapted more to the Republic of Korea. Therefore, this study aims to update the comprehensive effects of omega-3 consumption on CVD through a systematic review and meta-analysis of recently published articles from January 2000 to October 2022 and to estimate the potential savings in healthcare costs associated with omega-3 supplementation on CVD based on the Korean healthcare system’s perspective.

## 2. Materials and Methods

This study model utilized estimates of the net saving cost of omega-3 supplements in CVD healthcare coverage recipients; the potential cost offsets associated with CVD avoidance in members of the same group who could benefit from omega-3 fatty acid use consisted of two stages: (1) estimates were made of reductions in the relative risk of CVD associated with omega-3 intake from meta-analysis studies, with corresponding lower and upper 95% confidence intervals; (2) estimates were made of the economic benefit based on meta-analysis results using various data such as a target population, annual healthcare direct/indirect and lost productivity costs associated with CVD, and annual purchase expenses of omega-3 products in the Republic of Korea.

### 2.1. Systematic Review and Meta-Analysis

#### 2.1.1. Search Strategy

According to the Preferred Reporting Items for Systematic Reviews and Meta-analyses (PRISMA) guidelines [27], electronic searches were conducted in the Pubmed, Embase, and Cochrane Library databases published before 10 October 2022. The pre-defined keywords were used together using Boolean operators (“OR” and “AND”), including omega-3, cardiovascular disease, and a clinical trial with their alternate or subcategory words using Mesh terms. The detailed search queries are reported in Appendix A.

#### 2.1.2. Study Selection

In this study, the PICOS (population, intervention, comparison, outcomes, and study design) model for establishing the search strategy was set as follows: (P) all adults, regardless of sex, (I) omega-3 or its equivalent, (C) placebo, (O) cardiovascular events, and (S) randomized controlled trials (RCT). Study selection for meta-analysis was limited to English papers only, and the period was restricted from January 2000 to October 2023 to obtain the most recent data (See Appendix A). Studies were excluded if they were pre-print data or if they did not have clinical outcomes. Only one article was included in the review when 2 or more papers referred to the same study. Three authors (M.S.K, J.M.K, and S.K.L) assessed the first identified titles/abstracts for possible inclusion, and two authors (S.H.L and E.J.J) reviewed the full text against the inclusion criteria. Any disagreement regarding the study was evaluated by the third authors (B.K.L and W.Y.H).

#### 2.1.3. Data Extraction and Standardization

Duplicate studies were removed using EndNote (EndNote 20, Thomson Reuters, NY, USA), and studies that met the eligibility criteria were selected. Data for the meta-analysis were independently coded by two authors (J.M.K and J.Y.H) and tabulated using an Excel spreadsheet (Microsoft Excel 365, Microsoft Corp., Redmond, Washington, DC, USA). To extract the risk ratio (RR), calculation data and binary outcome data in cardiovascular disease were used. To transform the values, the median range, median IQR, and mean SE to mean SD values were applied using a website (mean variance and estimation (hkbu.edu.hk)) programmed based on Luo et al. [28], Wan et al. [29], and Shi et al. [30].

#### 2.1.4. Risk of Bias Assessment (RoB) and Publication Bias

Since all articles used in this study were RCTs, a risk of bias assessment was performed using the Cochrane Risk of Bias tool 2.0 (RoB 2.0) to assess the RCT developed by Cochrane. The RoB 2.0 tool uses five main categories (randomization process, deviations from intended interventions, missing outcome data, measurement of the outcome, and selection of the reported result) and their detailed criteria to ultimately differentiate the bias of a study as ‘low risk’, ‘some concerns’, or ‘high risk’ [31]. The assessment was performed independently by 2 reviewers (M.S.K and J.M.K), and disagreements during the evaluation were solved by discussing with the other researcher (J.Y.H) until a common opinion was reached. Funnel plots were used to evaluate publication bias.

#### 2.1.5. Statistical Analysis

To calculate the effect size of the studies included in the meta-analysis, the incidence of cardiovascular disease was analyzed by extracting the risk ratio (RR) and risk difference (RD) [32]. Random effects were applied to calculate the pooled effect estimates considering underlying variations across the included trials, and a forest plot with 95% effect estimates and confidence intervals (CI) was plotted. Heterogeneity was assessed using the *I*^2^ index and Q statistic with significant heterogeneity defined as *I*^2^ > 50.0%. Funnel plots and Egger’s regression were used to assess publication bias in the studies included in the meta-analysis. All data analyses were performed using the R statistical software version 4.2.1 (The R Foundation for Statistical Computing, Seoul, Republic of Korea), and *p* < 0.05 was considered statistically significant.

### 2.2. Estimating Social Economic Benefit

#### 2.2.1. Target Population

The target population included adults 50 years or older, classified as having major cardiovascular diseases (CVD) coded in the Health Insurance Review and Assessment Service Korea (HIRA), which is available via the HIRA big data open portal [33]. Based on the International Classification of Diseases and Korean Standard Classification of Diseases (KCD), CVD includes angina (I20), (acute) myocardial infarction and its complications (I21, I23), ischemic coronary syndromes (I24−I26), cardiomyopathy (I42, I43), heart failure (I50), and other heart disease that are not classified elsewhere (I51−I52).

#### 2.2.2. Cost Data and Source

Direct and indirect cost data were used for potential cost savings associated with a reduced risk of CVD, as described in Table 1. The direct medical cost was extracted from the outpatient and inpatient medical expenses used for the treatment of the disease in HIRA data. The direct non-medical cost was summed as transportation, caregiving, and leisure expenses for medical activities. Transportation costs resulting from hospital visits were estimated in a previous study [34] and calculated as an actual amount by applying the 2020 consumer price index. Caregiving expenses were calculated by multiplying the daily nursing cost by the number of hospitalizations using survey data from the Korea Consumer Agency [35]. Leisure cost was obtained from the national leisure survey from the Ministry of Culture, Sports and Tourism. Indirect costs associated with productivity loss and premature deaths were calculated using the average number of working days, the average wage, and employment rate by age group from the Korean government database, as described in Table 1 [36,37]. Productivity loss refers to the economic costs associated with lost workdays for hospitalized patients and lost work time due to outpatient care. Premature-death-related impairment of work capacity includes cases where the individual is unable to work at their previous level of productivity due to disease. The cost of premature death in this study was calculated as the product of the number of CVD patients multiplied by the employment rate, mortality rate, average annual wage, and duration of activity (5 years). The purchase prices for omega-3 intake from 2021 health functional food production data in the Korean Ministry of Food and Drug Safety (MFDS) for domestic products and the Import Food Market Information System (https://impfood.mfds.go.kr, accessed on 15 March 2023) for import products were used. The average prices were calculated by manually searching for sales on Google, Amazon, and NAVER for each. All costs were measured in Korean won (KRW) and were adjusted for inflation using the Korean consumer price index to reflect the 2021 KRW (USD 1 US equaled KRW 1260.75 as of June 2023). Detailed estimation equations are attached to Appendix A.

#### 2.2.3. Calculating Social Cost Saving

This study considered the total sum of direct and indirect costs as social costs and compared the gross cost estimates of omega-3 intake for the target population with the potential cost of the population who avoided CVD and who could benefit from omega-3. From the RR value derived after the meta-analysis, the absolute risk reduction (ARR), which represented the absolute difference in the risk of cardiovascular disease between the omega-3 intake group and the non-intake group, was calculated [41]. The number of people who could avoid CVD if all the target population consumed omega-3 was determined using ARR. Subsequently, the social cost calculated as the sum of direct and indirect costs was applied to this avoidance population to find the total social cost reduction. Since the target population was assumed to take omega-3, the purchase cost of omega-3 was applied to the entire target population. The net saving was obtained by subtracting the purchase cost of omega-3 from the total social cost savings.

## 3. Results

### 3.1. Study Selection

Figure 1 illustrates the procedure of the meta-analysis. As a result of searching articles, 2692 studies were initially identified from the three online databases. After deleting duplicates, reviewing the titles and abstracts, and assessing their eligibility through the whole text based on the inclusion and exclusion criteria of PICOS, 33 studies were finally included in the meta-analysis. Among them, GABA’s study considered two independent cases.

### 3.2. Characteristics of the Included Studies

The details of the research characteristics are summarized in Table 2. Most participants in each study had a history of heart disease, including myocardial infarction (MI), CHD, heart failure (HF), atrial fibrillation (AF), and diabetes, which could affect CVD, while eight studies (23.5%) targeted the general adult population rather than patients. The participant’s average age in each result ranged from 47.8 to 78.7 years, with all except one study considering participants aged 50 years and older. All subjects received omega-3 or equivalent interventions, and nine studies (10 cases) treated mono intervention as EPA or icosapent ethyl (IPE), while the rest used a combination with EPA, DHA, and other n3 polyunsaturated fatty acids (PUFA). The dosages in 24 studies (70.6%) used a consumption of 2 g or less, and 9 studies (10 cases, 29.4%) used a consumption of more than 2 g (2000 mg). The duration of the intervention ranged from 8 to 89 months, with 20 studies (58.8%) having a duration of 2 years or less and 13 studies (14 cases, 41.2%) having a duration of more than 2 years. CVD outcomes included the effects of omega-3 on integrated cardiovascular disease, referred to as MACE, total cardiovascular disease, or total coronary artery disease, depending on the researcher’s criteria and the specific cardiovascular disease outcomes assessed. When analyzing individual cardiovascular disease outcomes, only deaths attributed to cardiovascular disease were included, while deaths due to other causes were excluded from the study.

### 3.3. Risk of Bias Assessment

The results of assessing bias are presented in Figure 2. Due to the large number of long-term RCTs designed and conducted, most studies showed a low risk or some concerns. One study showed a high risk due to missing data bias as the number of participants decreased significantly compared to the initial study design despite a small sample size. Two studies [44] showed a tendency to interpret the study results excessively and diversely, indicating a high risk of selection for the reported result bias. In total, 10 studies had ‘some concern’ in measuring outcome bias. The overall bias assessment in all 33 studies was represented as ‘low risk of bias’. The RoB results of each study revealed that around five risks of bias, domains were added in Appendix A.

### 3.4. Results of Meta-Analysis and Funnel Plot

Figure 3 shows a forest plot of RR and the risk difference (RD) for the main outcome. A total of 31 outcomes from 30 studies involving 160,677 participants (80,426 in the intervention group and 80,251 in the control group) were extracted to investigate the effect of omega-3 intake on all cardiovascular diseases. The results indicated a significant reduction in cardiovascular diseases with omega-3 intervention (RR = 0.92, 95% CI: 0.86~0.97, *I*^2^ = 77%, Q = 129.72, *p* < 0.01] (Figure 3a). The absolute risk reduction (ARR) was determined from the absolute value of RD’s function value using the R program (RD = −0.0148, 95% CI: −0.0253 to −0.0043, *I*^2^ = 76%, Q = 124.24, *p* < 0.01] (Figure 3b) The ARR value was 0.0148 (1.48%), which meant about 67.6 individuals needed to take omega-3 to treat one CVD person.

Since heterogeneity was over 50%, subgroup analysis was performed based on the difference in duration (2 years or less vs. more than 2 years) and dosage (2 g or less vs. more than 2 g) of omega-3 intake, as seen in previous studies [69,70] (see Appendix A). The reduction effect of cardiovascular disease decreased significantly when the intake dosage exceeded 2 g (RR = 0.82, 95% CI: 0.74~0.91, *I*^2^ = 74%, *p* < 0.01) compared to 2 g or less (RR = 0.97, 95% CI: 0.94~1.01, *I*^2^ = 52%, *p* < 0.01). In terms of intake duration, the standard was set to 2 years (48 months), with 2 years or less (RR = 0.98; 95% CI: 0.90~1.07, *I*^2^ = 52%, *p* < 0.01) compared to over 2 years (RR = 0.88; 95% CI: 0.82~0.95, *I*^2^ = 86%, *p* < 0.01). The effect of reducing the risk of cardiovascular disease was more significant when the intake duration was over 2 years. However, the cost-saving analysis in this study did not use the subgroup results.

**Figure 3 healthcare-11-02365-f003:**
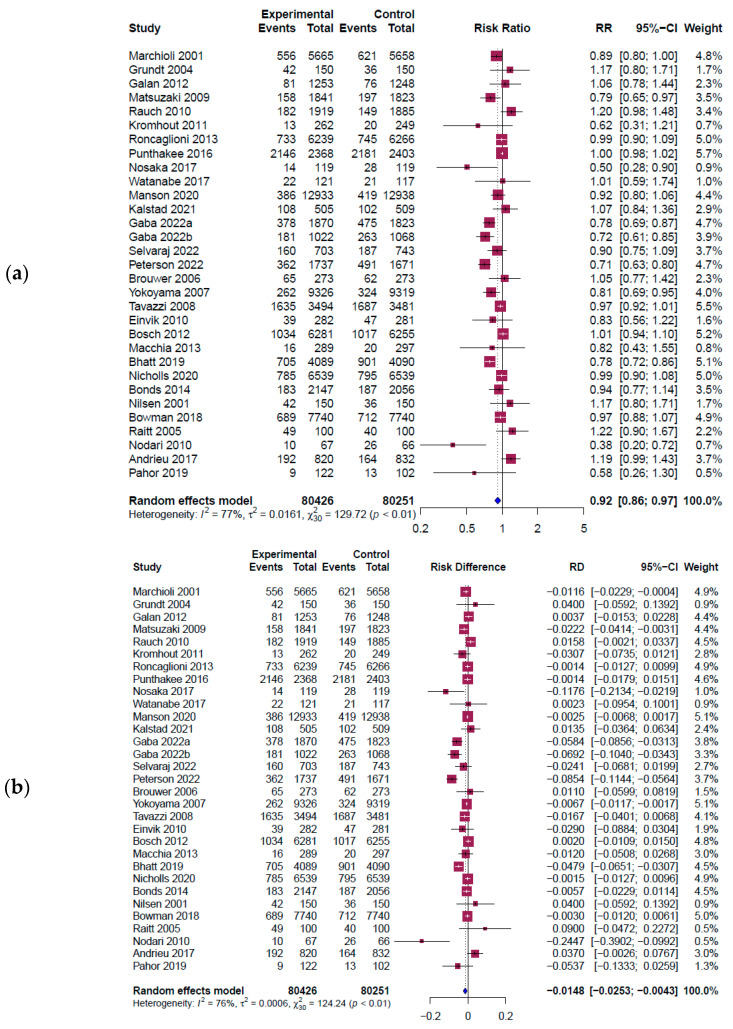
Forest plot results from a meta-analysis [8,10,11,42,43,46,47,48,49,50,51,52,53,54,55,58,59,60,61,62,63,64,65,66,67,68,71]. The figure shows the pooled estimate of (**a**) RR results; (**b**) RD results. Experimental: omega-3 intervention group; event: CVD incidence; CI: confidence interval; RR: risk ratio; RD: risk difference; blue diamond symbol (◊): overall effect size; red square symbol (□): effect size of individual studies; dashed line: 95% confidence interval for individual studies.

Funnel plots to evaluate the publication bias are shown in Figure 4. Black dots represent RR in all the studies, respectively. To mitigate the publication bias observed asymmetrically on the plot, six additional studies are represented by white dots. Egger’s regression test was conducted to identify the publication bias, which can be observed asymmetrically on the plot. As the *p*-value was 0.0254 (not shown), the articles included in this meta-analysis were concluded to be symmetrical with no publication errors.

### 3.5. Estimation of Cost Savings

Table 3 illustrates the population and calculation results that estimate the effect of social cost savings in omega-3 supplements on CVD based on various data related to Korean healthcare costs in 2018–2021. As of 2021, the CVD population over 50 was 1,160,730, which corresponds to 5.4% of the total population over 50 and 95.4% of the CVD population of all ages. It indicates that domestic CVD patients were almost always 50 or older. Reflecting the ARR value (1.48%) calculated in the meta-analysis, the population (about 17,179 people) that could avoid cardiovascular disease was calculated where the entire omega-3 supplement intake group (1,160,730 people) was assumed in 2021. For CVD-related social costs, direct medical/non-medical and indirect costs were calculated by year using the above-described corresponding data and were then converted into social costs per capita. When calculating the indirect cost of premature death, the average mortality rate (34.4%) was calculated from the mortality rates due to CVD, as reported in previous studies [39,40]. The total social cost was obtained by multiplying the social cost per capita and the target CVD population. In the case of 2021, it was about KRW 43,307 billion. Social cost savings were about KRW 641 billion (about 507 million USD) in 2021 by multiplying the social cost per capita and the avoided CVD population.

The sales prices of 440 domestic and 276 imported omega-3 supplement products were investigated, and the price per gram and daily intake were calculated. The daily purchase expense was obtained by multiplying the daily intake (d/day) and the average selling price (KRW/g). After converting into a weighted average of domestic and imported purchase prices, the annual average purchase cost of omega-3 supplements was KRW 293,918 per capita in 2021. Since it was assumed that the entire target population would purchase omega-3 supplements, the total purchase expense of omega-3 was about KRW 341 billion (about 270 million USD) in 2021 by multiplying the product of the purchase cost per person and the number of CVD subjects aged 50 years or older.

By subtracting the purchase expense of omega-3 supplements from the total social cost savings, the net social cost saving was about KRW 300 billion (about USD 237 million) in 2021. The ratio of the social cost saving (Benefit) to the omega-3 purchases cost (Cost), called the BC ratio, ranged from 1.55 to 1.88 in 2018–2021, which could be interpreted as an effect or benefit of social cost reduction from the intake of omega-3 supplements. However, although the omega-3 purchase expenses used in this study were from the government data informed via MFDS, there might be errors as the consumption price was applied only in 2021 as the result of the survey. Further studies are needed to secure or investigate data on consumption prices by year.

## 4. Discussion

This study tried to adopt a similar method by evaluating the social and economic effects of healthy functional foods studied in the US, Australia, and Europe [22,23,24]. Two points of this study are whether omega-3 intake effectively reduces the incidence of CVD using recently published RCT studies and whether its effects can be applied to domestic adults to reduce social costs, including healthcare costs. There was an attempt to estimate the cost-saving effect of omega-3 intake on healthcare in the Republic of Korea in 2015 [26], where some issues were discussed that needed addressing, and updated costs were used to estimate their study. In our study, therefore, we conducted a comprehensive meta-analysis using the most updated studies possible, then linked the meta-analysis results with various data relating to Korean adults’ healthcare costs, including direct medical/non-medical costs and indirect costs, to evaluate the health and economic benefits of omega-3 supplements.

Our meta-analysis results showed that omega-3 intake could significantly reduce the occurrence risk of overall cardiovascular disease by approximately 8% (RR = 0.92, 95% CI: 0.86~0.97) and the ARR was 1.48%, which corresponds to the incidence difference between the experimental and control groups. In other words, 1.48 out of 100 people in the omega-3 intake group could avoid CVD. This study calculated the ARR value from the RR value derived through meta-analysis. The number of people who could avoid CVD was calculated, assuming that domestic patients with omega-3 intake consumed all omega-3 supplements. A recent comprehensive meta-analysis study on CVD published in 2021 [72] showed no significant effect of omega-3 intake on reducing overall CVD risk, which differs from the results of this study. Although the study in 2021 included 39 documents, it had a smaller sample size than ours due to differences in the size of the recruitment population in the individual documents for their meta-analysis. Although meta-analyses have been evaluated as having strong scientific evidence, they can show different results. In particular, differences in results often occur due to different sample sizes, like the difference between the 2021 study and our study, and relatively large deviations can also occur in smaller sample studies. A study published in 2019 [73], which used only 12 relatively fewer documents, showed a similar trend to ours, with all 12 studies being large-scale randomized controlled trials (RCTs).

Additionally, the subgroup analysis results in our study indicate that higher doses and the long-term consumption of omega-3 could further enhance these effects. Regarding duration or dosages, a significant reduction in CVD was observed when 2 g of omega-3 was consumed for over 2 years, indicating the potential for disease prevention through long-term use or higher intakes. However, safety studies on consuming large amounts of omega-3 over a long period are needed. Additionally, six of the individual studies included in this meta-analysis conducted omega-3 intake studies on general adults, not populations who previously experienced cardiovascular disease. Although significant results were not obtained in the meta-analysis, individual results mostly showed a decrease in cardiovascular disease risk, suggesting that omega-3 intake can help prevent the onset of cardiovascular disease. Therefore, further research is needed to obtain significant results through studies on the characteristics of the general population and additional omega-3 intake studies rather than just on patient populations. Based on our results, the health benefits of omega-3 on CVD have been proven, especially when taken over a long time, like dietary supplements or healthy functional foods, which could be more beneficial.

Based on our meta-analysis results, we derived health and economic benefits from the omega-3 intake in adults over 50 years of age in the Republic of Korea and calculated the social cost savings to be about KRW 300 billion as of 2021. This estimates a total saving of KRW 13,874 per person over 50 years of age and KRW 258,458 per CVD person over 50 in 2021. Additionally, several assumptions were considered when the meta-analysis results were linked with the domestic data. In the population, we selected Korean adults over 50 years of age and CVD patients among them, considering the meta-analysis population and the age group presented in the CRN report [24]. Assuming that all CVD subjects over 50 in the Republic of Korea consume omega-3, potentially avoided CVD was calculated by applying the ARR value. Finally, it was assumed that all CVD subjects might purchase omega-3 to calculate the purchase cost. People without diseases also purchase nutritional supplements such as omega-3, but this study limited consumers and purchasers to CVD subjects and applied them conservatively.

In a previous study, the potential net savings in avoided CHD-related hospital utilization costs after accounting for the cost of omega-3 dietary supplements at preventive daily intake levels was more than USD 3.88 billion in cumulative healthcare cost savings from 2013 to 2020 in the US [74]. In a following study in 2022, it was expected that approximately USD 3.41 billion of the USD 3.70 billion in net potential direct savings from avoided coronary artery disease (CAD) hospitalization events was saved. Additionally, it was suggested that the target high-risk population could realize significant cost savings from using omega-3 EPA+DHA dietary supplements [24]. For Australia, treatment with icosapent ethyl was considered, as omega-3 was associated with both higher costs and benefits using the method based on the quality-adjusted life year (QALY) and life year (LY), resulting in an incremental cost-effectiveness ratio of AUD59,036/QALY or AUD54,358/LY [22]. The above two countries’ research had a similar purpose to our research, but their methods were different. Various methods are available to estimate potential health cost savings, including cost-effectiveness, cost–benefit, and cost-of-illness analyses. The model selection needs to be based not only on the aim of the study but also on available data sources. As described above, we used various direct/indirect data, especially the income and work loss cost due to premature death. We also estimated the expected purchase cost of consumer products by referring to domestic 2021 health functional food production data and import status. However, it is still necessary to conduct large-scale RCT studies targeting Koreans to consider domestic and international demographic characteristics in more detail.

Therefore, this study suggests a method to estimate the social cost savings effect of omega-3 on CVD in the Republic of Korea using stronger evidence from the meta-analysis and provides a basis for establishing national policies on the intake of healthy functional foods such as omega-3.

## 5. Limitation

This study has some limitations. First, Since the diseases included in cardiovascular disease might differ in each nation, the CVD of this study could be interpreted differently. This study used 11 diseases categorized as ‘national interest diseases’, including many cardiovascular diseases in the Republic of Korea. Second, the literature subjects in this study’s meta-analysis included both general adults and patients with various diseases. It is necessary to classify and analyze the subjects through further study. Additionally, this study contains references through October 2022; therefore, we should consider making updates regarding further studies published in 2023. Third, the articles (except 4) used in this meta-analysis needed more available safety outcomes. Thus, this study did not provide an overall evaluation of safety outcomes. Lastly, the discount price did not apply when investigating the domestic and international sales prices of omega-3, so there was a tendency to overestimate the prices.

## 6. Conclusions

The reliability of economic assessments depends on accurate clinical benefit assessments and cost estimates. This study included some uncertainty elements, but the latest meta-analysis results and domestic data were applied conservatively. Based on the results of this study, omega-3 supplementation among Korean adults suggests the potential to reduce cardiovascular disease risk and social costs. Furthermore, this study may provide valuable information or a methodology to identify health and economic benefits arising from omega-3 as well as healthy functional foods and other supplements when planning health policies in the Republic of Korea.

## Figures and Tables

**Figure 1 healthcare-11-02365-f001:**
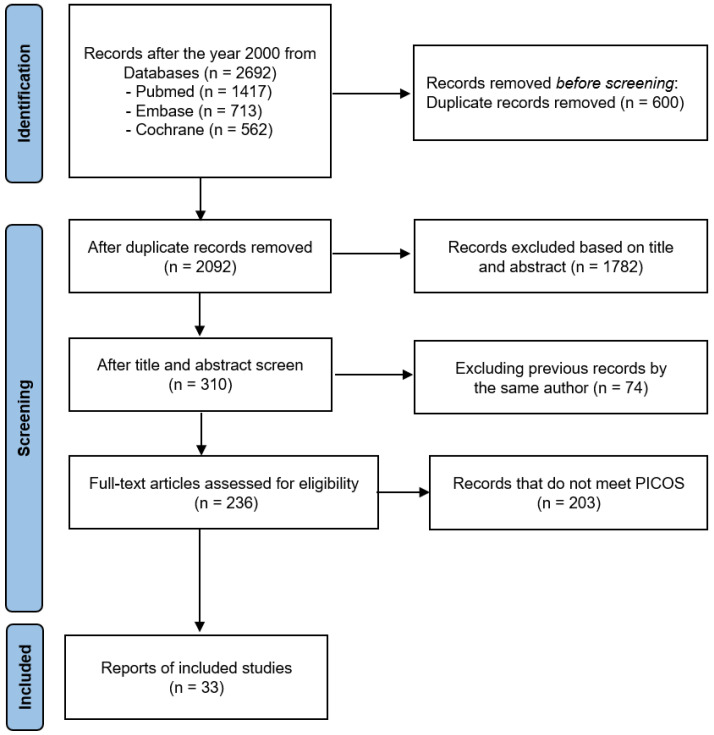
The PRISMA 2020 flow diagram for this meta-analysis.

**Figure 2 healthcare-11-02365-f002:**
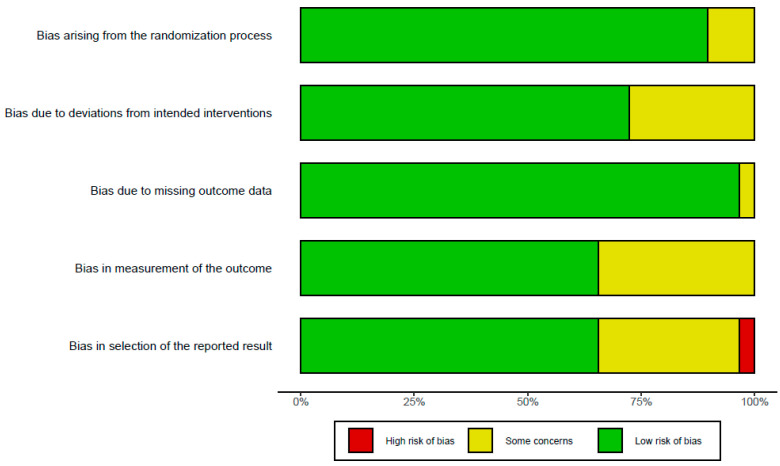
Risk of bias assessment of included studies using five domains in RoB 2.0 criteria.

**Figure 4 healthcare-11-02365-f004:**
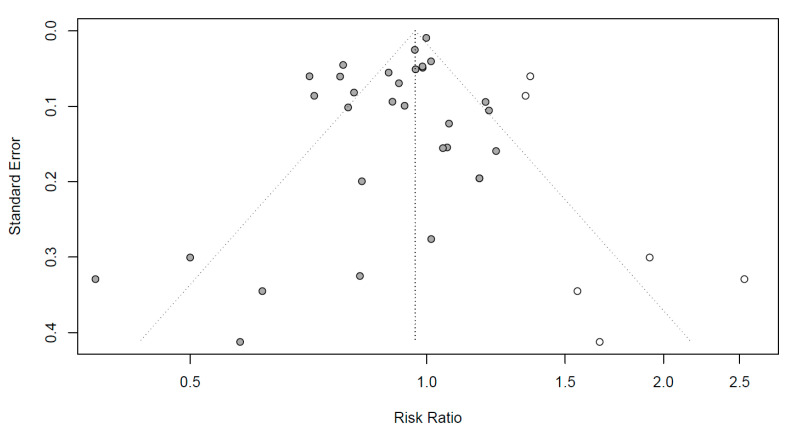
Funnel plots for publication bias evaluation. X-axis: effect size; y-axis: standard error; black dots: individual studies; white dots: dots added to be symmetry; diagonal dotted line: 95% confidence interval; vertical dotted line: overall effect size.

**Table 1 healthcare-11-02365-t001:** List of economic data, their source, and equation.

Category			Source	Equation
Direct cost	Medical	Outpatient treatment	Health Insurance Review and Assessment Service Korea (HIRA) [33]	
		Inpatient treatment		
	Non-Medical	Transportation	Park et al., 2000 [34]	Average daily transportation cost × Outpatient treatment day
		Caregiving	A survey on the use of patient care intermediary services, 2022 [35]	Average daily caregiving cost × Inpatient (hospitalization) day
		Leisure	A national leisure survey [36]	Average daily leisure cost × Inpatient (hospitalization) day
Indirect cost		Productivity Loss	Economic Activity Population Survey [37] Employment and Labor Statistics, 2020 [38] Statistics on Causes of Death [37] Cheon et al. 2016 [39] Asan Medical Center [40]	Number of CVD patients in population × employment rate × average daily wage× [hospitalized treatment days per capita + (outpatient treatment days per capita ÷ 3)]
		Loss due to premature death		Number of CVD patients in population × employment rate × mortality rate × average annual wage × period of activity (5 years)

**Table 2 healthcare-11-02365-t002:** Characteristics of all studies used in meta-analysis.

First Author (Year)	Population	Age	Intervention	Dosage (mg)	Duration (Month)	CVD Outcome
Marchioli (2001) [40]	MI	(T) 59.3 ± 10.6	EPA+DHA	866	42	Death, MI, stroke, Revascularization
Nilsen (2001) [42]	MI	(I) 63.8 ± 11.0 (C) 63.2 ± 11.1	EPA+DHA	4000	18	Death, MI, Revascularization, Angina, Resuscitation
Grundt (2004) [43]	MI	(I) 67.2 ± 15.7 (C) 66.3 ± 13.1	EPA+DHA	866	18	Death, MI, Resuscitation, Angina, Revascularization
Leaf (2005) [44]	ICD	(I) 65.7 ± 11.6 (C) 65.3 ± 11.7	EPA+DHA	2600	12	Death, ICD, HF
Raitt (2005) [45]	AF	(I) 63 ± 13 (C) 62 ± 13	EPA+DHA	1800	24	Death, MI, Angina, Arrhythmia, Revascularization
Brouwer (2006) [46]	AF	(I) 60.5 ± 12.8 (C) 62.4 ± 11.4	EPA+DHA + other n3 PUFA	957	12	Angina, HF, Arrhythmia
Yokoyama (2007) [11]	All adult	(I) 61 ± 8 (C) 61 ± 9	EPA	1800	55	Death, MI, Angina Revascularization
Tavazzi (2008) [9]	HF	(T) 67 ± 11	EPA+DHA	866	54	Death, MI, HF, Stroke
Matsuzaki (2009) [47]	CHD	(T) 63 ± 8	EPA	1800	55	Death, MI, Angina, Revascularization
Nodari (2010) [48]	MI	(I) 61 ± 11 (C) 64 ± 9	EPA+DHA	866	12	HF
Einvik (2010) [49]	All adult	(I) 70.4 ± 2.9 (C) 69.7 ± 3.0	EPA+DHA	2400	36	Death, MI, Stoke, Revascularization
Galan (2010) [15]	CHD	(I) 61.1 ± 11.1 (C) 61.7 ± 9.7	EPA+DHA	600	56	Death, MI, Stroke, Revascularization
Rauch (2010) [50]	MI	(I) 63.3 ± 13.4 (C) 63.3 ± 13.4	EPA+DHA	840	12	Death, Revascularization, ICD
Kromhout (2011) [51]	MI	(I) 69.4 ± 5.7 (C) 69.3 ± 5.7	EPA+DHA	400	40	Death, Arrhythmia
Bosch (2012) [52]	diabetes	(I) 63.5 ± 7.8 (C) 63.6 ± 7.9	EPA+DHA	900	74	Death, MI, Stroke, HF, Angina, Revascularization
Macchia (2013) [53]	AF	(I) 66.3 ± 12.0 (C) 65.9 ± 10.5	EPA+DHA	866	12	Death, MI, Stroke, HF, AF
Roncaglioni (2013) [54]	CVD	(I) 63.9 ± 9.3 (C) 64.0 ± 9.6	EPA+DHA	850	36	Death, MI, Stroke, Arrhythmia
Bonds (2014) [55]	All adult	(I) 74.6 ± 11.1 (C) 74.0 ± 11.1	EPA+DHA	1000	58	Death, MI, Stroke, Angina, HF, Revascularization
Wilbring (2014) [56]	MI	(I) 67.6 ± 10.9 (C) 67.5 ± 7.9	EPA+DHA + other n3 PUFA	1800	27	Death, AF, MI
Sanyal (2014) [57]	NASH	(I) 47.8 ± 11.1 (C) 50.5 ± 12.5	EPA	1800	12	Angina
Punthakee (2016) [58]	MI	(I) 63.0 ± 7.4 (C) 63.3 ± 7.6	EPA+DHA	1000	74	Death, MI, Stroke, arrhythmia, HF, Revascularization, Angina
Nosaka (2017) [59]	CHD	(I) 70 ± 11 (C) 71 ± 12	EPA	1800	12	Death, MI, Stroke, HF, Revascularization
Watanabe (2017) [60]	CHD	(I) 67 ± 10 (C) 68 ± 10	EPA	1800	8	Death, MI, Angina, Stroke, Revascularization
Andrieu (2017) [61]	All adult	(I) 75.4 ± 4.4 (C) 75.6 ± 4.7	EPA+DHA	2000	36	CVD
Bowman (2018) [62]	diabetes	(I) 63.3 ± 9.2 (C) 63.6 ± 9.2	EPA+DHA	840	89	Death, MI, Stroke, Revascularization
Bhatt (2019) [8]	CVD or diabetes	(I) 63.3 ± 8.9 (C) 63.3 ± 8.9	IPE	4000	59	Death, MI, Stroke, Angina, Revascularization
Pahor (2019) [63]	All adult	(I) 78.0 ± 5.6 (C) 77.4 ± 5.3	EPA+DHA	2100	12	CVD
Nicholls (2020) [10]	All adult	(I) 62.5 ± 9.0 (C) 62.5 ± 9.0	EPA+DHA	4000	54	Death, MI, Stroke, Angina, Revascularization
Manson (2020) [64]	All adult	(I) 67.2 ± 7.1 (C) 67.2 ± 7.1	EPA+DHA	1000	64	Death, MI, Stroke, Revascularization
Kalstad (2021) [65]	MI	(I) 74.7 ± 4.5 (C) 74.7 ± 4.5	EPA+DHA	1590	23	Death, MI, Stroke, HF, AF, Revascularization
Gaba (2022) [66]	MI	NC	IPE	4000	58	Death, MI, Angina, Stroke, Revascularization
	All adult	(I) 62.6 ± 8.67 (C) 62.5 ± 8.70	IPE	4000	58	
Selvaraj (2022) [67]	HF	(M) 63.0	IPE	4000	24	Death, MI, Stroke, Angina, Revascularization
Peterson (2022) [68]	PCI	(I) 63 ± 8.9 (C) 62.6 ± 9.6	IPE	4000	58	Death, MI, Stroke, Angina, Revascularization

MI, myocardial infarction; ICD, implantable cardioverter defibrillator; CHD, coronary heart disease; CVD, cardiovascular disease; HF, heart failure; AF; atrial fibrillation; EPA, eicosapentaenic acid; DHA, docosahexaenoic acid; IPE, icosapent ethyl; n3 PUFA, omega-3 poly unsaturated fatty acid; PCI, Percutaneous Coronary Intervention; (I) intervention group age; (C) control group age; (T) total participant age; (M) median; NC, not clear.

**Table 3 healthcare-11-02365-t003:** Results of cost saving estimators in 2018–2021.

Metrics		2018	2019	2020	2021
Total population over the age of 50 (N)		19,809,141	20,459,327	21,091,560	21,622,993
CVD population (N)		1,103,620	1,147,646	1,153,997	1,217,044
CVD population over the age of 50 (N)		1,037,275	1,086,646	1,102,353	1,160,730
Absolute risk reduction (ARR, %) ^1^		1.48			
Population that can avoid CVD (N) ^2^		15,352.67	16,082.36	16,314.82	17,178.80
Direct medical cost (KRW billion) ^3^		1154.9	1250.4	1257.9	1342.3
Direct non-medical cost (billion KRW)	Transportation	59.5	58.7	57.7	64.0
	Caregiving	172.5	171.6	160.9	155.4
	Leisure	8.1	8.3	7.7	7.1
Indirect cost (billion KRW)	Productivity Loss	152.4	160.5	153.0	155.8
	Loss due to premature death ^4^	37,012.0	40,231.3	40,555.4	43,687.7
Social cost per person (KRW million)		31.68	34.41	34.67	37.31
Sum of social cost with CVD (KRW billion) ^5^		32,861	37,391	38,219	43,307
Sum of social cost saving (KRW billion) ^6^		486	553	566	641
(USD million) ^9^		385	438	449	507
Omega-3 supplement purchase expense					
	One person/year (KRW)	310,571	328,007	311,600	293,918
	CVD population/year (KRW billion) ^7^	322	356	344	341
	(USD million)	255	282	273	270
Net social cost saving (KRW billion) ^8^		164	197	222	300
(USD million)		130	156	176	237
Social cost saving/omega-3 purchase expense		1.88	1.55	1.65	1.88

CVD, cardiovascular disease; ARR, absolute risk reduction; KRW, Korean dollar won; USD, US dollar. ^1^ Absolute value of RD by calculating from the meta-analysis’s risk ratio (RR). ^2^ By multiplying the CVD population over the age of 50 and ARR. ^3^ Sum of outpatient and inpatient treatment costs for CVD from Health Insurance Review and Assessment Service Korea (HIRA) data. ^4^ By multiplying CVD population, employment rate, mortality rate, average annual wage, and period of activity (5 years), where the average mortality rate due to CVD was referred to Cheon et al., 2016 with Asan medical center information [39,40]. ^5^ By multiplying the CVD population over the age of 50 and the social cost per person. ^6^ By multiplying the population that can avoid CVD over the age of 50 and the social cost per capita. ^7^ By multiplying the CVD population over the age of 50 and the annual purchase expense of omega-3 supplements per capita. ^8^ by subtracting the purchase expenses of omega-3 supplements from the total social cost saving. ^9^ by exchanging USD 1 for KRW 1260.75 as of June 2023.

## Data Availability

Not applicable.

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
