# Peer review of "Assessing Health and Economic Benefits of Omega-3 Fatty Acid Supplementation on Cardiovascular Disease in the Republic of Korea"

_healthcare, 2023, doi:10.3390/healthcare11162365_

Round 1
Reviewer 1 Report
This meta-analysis assessed the health and economic benefits of omega-3 associated with cardiovascular disease (CVD). The authors showed a significant reduction of omega-3 in CVD and estimated that omega-3 consumption could result in an economic benefit of 300 billion KRW by subtracting the purchase expenses of omega-3 supplements from the total social cost saving.
The article is relevant as it highlights the importance of omega-3 in CVD.
Some advice for authors:
- The meta-analysis includes observational studies and randomised trials (RCTs). I would indicate how many studies are retrospective, prospective and RCTs.
- Since many studies are observational, I would emphasize this in the limitations.
- For the risk of bias of RCTs, you should use the Cochrane Risk of Bias tool for Randomised Controlled Trials and not the RoB 2.0.
- Studies have different populations (MI, AF, diabetes...), a subgroup analysis in patients with MI could give an important message as well as reduce heterogeneity.
- Safety outcomes were not evaluated. If data were available, I would recommend doing so. Alternatively, I would point out that their absence is a major limitation of the study.
- Line 127 correct “RRisma”
Author Response
Dear Reviewer 1
Please see the attachment.

Reviewer 2 Report
Thank you for the opportunity to review the manuscript. The authors analyzed the benefits of omega-3 fatty acid supplementation in the Korean cardiovascular context. I would like to present my suggestions.
Abstract
The authors could write the date of the study.
The objectives could be better described as in lines 108-112.
Introduction
What is “MI”? (Line 65). The abbreviation MI (myocardial infarction) is only described in line 248.
Materials and Methods
2.1.2. Study selection
For the authors, what is the age range included in "all adults"? (Line 136). I ask this because including studies with younger adults taking supplementation and followed for a short time could lead to a false conclusion of positive benefits.
I suggest filling in the information about the date of the study: ...English papers only, and the period was restricted from 2000….” (Line 139). From 2000 to 2022? This information is only available as “Table S1”.
Results
3.2 Characteristics of the included studies
The authors included 33 studies (Figure 1 and Table 2). I did not understand “…Dosages resulted in 24 studies (70.6%) with 253 a consumption of 2 g or less and ten studies (29.4%) with a consumption of more than 2 g 254 (2,000mg). Duration of intervention ranged from 8 to 89 months, with 20 studies (58.8%) 255 having a duration of less than 2 years and 14 studies (41.2%) having a duration of more 256 than 2 years.” (Lines 253-257). Are there 33 or 34 studies?
3.4. Results of meta-analysis and funnel plot
“The reduction effect of cardiovascular disease was decreased significantly when the intake dosage exceeded 2g (RR = 0.82, 95% CI: 0.74 ~ 0.91, I 2 = 74%, P < 0.01) compared to 2g or less (RR = 0.97, 95% CI: 0.94 ~ 1.01, I 2 = 52%, p < 0.01).” (Lines 297-299) I would like to know in which age group the authors verified these results. I ask this because including studies with younger adults taking supplementation and followed for a short time could lead to a false conclusion of benefits.
3.6. Estimation of cost savings
I suggest adding information in US dollars as in the introduction (text and table3)
Limitation
The inclusion of adults of all ages (Line 136) may lead to false conclusions about the benefits of supplementation. Could the authors provide the age ranges of participants in each study? Did the included studies actually have young adults?
References
The authors could include some reference to the year 2023.
Author Response
Dear Reviewer 2
Please see the attachment.

Round 2
Reviewer 1 Report
Congratulations to the authors